# Isolated Progression of Multiple Myeloma into the Extramedullary Plasmacytoma of Dura Mater: A Case Report and Review of the Literature

**DOI:** 10.3390/biomedicines11041225

**Published:** 2023-04-20

**Authors:** Agata Tyczyńska, Mikołaj Turski, Ewa Zarzycka, Jan Maciej Zaucha

**Affiliations:** 1Department of Hematology and Transplantology, Medical University of Gdańsk, 80-214 Gdańsk, Poland; 2Student Scientific Circle at the Department of Hematology and Transplantology, Medical University of Gdańsk, 80-210 Gdańsk, Poland

**Keywords:** multiple myeloma, central nervous system plasmacytoma, extraosseous plasmacytoma, plasmacytoma

## Abstract

Multiple myeloma (MM) is a disease caused by the uncontrolled proliferation of clonal plasma cells in bone marrow. Extramedullary plasma cell infiltrations may occur at the time of diagnosis but usually arise during systemic disease progression. Central nervous system (CNS) plasmacytomas are extremely rare (less than 1% of patients with MM) and usually occur as a result of systemic disease progression. The frequency of extramedullary progression to CNS without simultaneous systemic progression is not known. Here, we present a challenging case in which local disease progression to CNS occurred without any signs of systemic progression. The extramedullary plasmacytoma originated from the dura mater of the brain mimicking a brain tumor. We review and discuss further treatment options that are available in such rare clinical scenarios in relation to the treatment already undertaken.

## 1. Introduction

Multiple myeloma (MM) cells predominantly infiltrate bone marrow. In about 5–7% of cases, extramedullary lesions are found at diagnosis [1], which occur through the infiltration of soft tissues adhering to the affected bone cortex or, in some cases, also through hematogenous spread. Central nervous system (CNS) infiltration by plasma cells at diagnosis is a rather rare phenomenon, normally representing only 1–4% of cases [2] but recently rising to almost 12%, likely due to increased detection due to the use of modern imaging techniques [3]. Extramedullary infiltrations usually occur during systemic disease progression and are indicative of poor prognosis. It is reported that CNS involvement in MM is more common in younger patients (50–60-year-old) compared with the median age of 70 years for diagnosis of MM [1,4,5,6], regardless of gender [6]. Central nervous system involvement in patients with MM may be challenging due to heterogeneous symptoms that can be confounded by neurological symptoms caused by the typical features of myeloma or treatment side effects. In some cases, the appearance of extraosseous lesions can occur without any other signs of systemic progression. Plasmacytoma of the CNS in a patient with a stable MM may pose an even greater clinical challenge. Symptoms are not specific and mimic those of other more common tumors in this location, such as meningioma or lymphoma [7,8], and it is unclear how such patients should be treated. Here, we present a patient with MM considered have achieved very good partial remission who developed CNS plasmacytoma without any other symptoms or laboratory evidence of systemic MM progression.

## 2. Case Report

A patient working as a car mechanic presented with spinal aches lasting for about two years. Symptoms were attributed to his work conditions. In addition, he was treated with antibiotics several times due to recurrent airway and urinary tract infections, and immunodeficiency was suspected. The bone marrow biopsy performed during the work-up revealed 80% infiltration of bone marrow with clonal plasma cells producing monoclonal IgG lambda protein (serum concentration of 48 g/L). He did not meet any CRAB criteria: there were no bone lytic lesions nor hypercalcemia (8.8 mg/dL), renal failure (creatinine 0.79 mg/dL, GFR > 90 mL/min/1.73 m^2^), or anemia (Hb 12.1 mg/dL). However, he met two SLIM criteria (infiltration of bone marrow more than 60% of plasma cells and kappa/lambda index < 0.01) and started VTD (bortezomib, thalidomide, and dexamethasone) therapy. Unfortunately, no cytogenetic tests were performed at the time of diagnosis. After 5 cycles, he achieved a very good partial response (VGPR; monoclonal protein MP = 4.08 g/L). The treatment was complicated by right leg thrombosis (despite the use of antithrombotic prophylaxis) and visceral and grade 1 peripheral neuropathy. As a result, thalidomide therapy was discontinued, and administering of the bortezomib dose was reduced to once a week. The patient was awaiting autologous hematopoietic cell transplantation (auto-HCT). After the 7th VTD cycle (December 2017), biochemical disease progression was documented (an increase in MP levels to 9.1 g/L). In January 2018, the second line of treatment was started (bendamustine) followed, one month later, by the first auto-HCT after typical conditioning using melphalan (200 mg/m^2^). The procedure was complicated by viral pneumonia (flu A1H1). After transplantation, VGPR was achieved with MP concentration around 1.1 g/L. No maintenance treatment was administered due to a lack of reimbursement. From March 2018 onward, the patient’s MP levels remained stable, at 1.2–2.0 g/L, without any other symptoms of active disease.

In October 2020, the patient was admitted to the emergency department after having experienced mostly left-sided headaches for over a month, right upper limb muscle weakness, and dizziness. Besides pre-existing polyneuropathy, the neurological examination also revealed right-sided pyramidal syndrome. A computed tomography scan of the head showed an extracerebral focal lesion (24 mm × 48 mm × 66 mm) located in the left frontoparietal area and pressing into the adjacent brain tissue. No bones of the skull were infiltrated. Magnetic resonance imaging (MRI) of the head (Figure 1) confirmed the presence of a tumor within the dura mater. Meningioma was suspected, and the patient qualified for nonradical tumor resection.

Histopathological examination revealed the presence of monoclonal plasma cells. As a consequence, the patient was evaluated for systemic myeloma progression. The bone marrow biopsy showed 4% plasma cells without the restriction of chain expression. In the next generation, flow cytometry plasma cells constituted 0.6% of all nucleated cells, out of which only 0.13% were clonal. The MP concentration was stable at 2.56 g/L. Flow cytometry of CNS fluid did not reveal plasma cells. In bone tomography, no evidence of osteolysis was documented. In February 2021, the patient underwent radical radiotherapy of the tumor bed of 40 Gy divided into 20 fractions and boosted up to 50 Gy for residual tumor. After completion of radiotherapy in May 2021, the second auto-HCT was performed followed by lenalidomide for maintenance. The patient was conditioned with two drugs that cross the blood– brain barrier (BBB), carmustine and melphalan, in two days [9,10]. Currently, the patient remains in good physical and mental condition without evidence of CNS relapse and with a stable MP concentration, of about 3 g/L, and negative positron emission tomography findings.

## 3. Discussion

The isolated intracranial progression of MM without any simultaneous signs of systemic progression is a very rare occurrence, and the exact incidence of this complication is unknown. The younger age of our case patient (44 years) is in line with literature reports of the higher tendency of CNS involvement in younger patients (less than 50–60 years old) [1,6,11] Plasmacytoma of CNS often imitates other lesions that occur more frequently in this location. Imaging examination might be misleading and is not sufficient as a basis for making a definite diagnosis [2,12,13]. Diagnosis is not straightforward and requires histopathological verification. In our case, the intracranial lesion initially was not suspected to be related to MM and considered to be a meningioma. It is suggested in the literature that MRI with contrast is more sensitive than computed tomography [14,15,16], but false negatives are still reported in 10% of cases [17]. Our patient was assessed with both methods, and neither allowed us to even suspect tumor histopathology. Tumor biopsy remains the gold standard for definitive diagnosis. Cerebrospinal fluid (CSF) examination might be helpful in cases where monoclonal plasma cells are detected in the CSF [18], especially in cases where performing tumor biopsy is difficult. However, in the case of dura mater involvement, plasma cells might be absent from CSF [19], as in our case.

Most of the reported cases are either with CNS involvement during the systemic progression of MM [16] and few in which CNS involvement preceded [20,21,22,23,24] or prompted [25] the diagnosis of MM. Jurczyszyn et al. described the biggest cohort to date of 172 patients diagnosed during 1995–2014 with histologically or radiologically proven CNS plasmacytomas that were non-continuous with bones. In 22% of these patients, CNS infiltration was documented at diagnosis, while in the others it was during the course of the disease at the median time of 25 months since diagnosis. Information about how many of these patients had no systemic progression is unfortunately missing. We have searched the literature to identify patients similar to our case. Only Yi Chen et al. described a case of a patient with disease progression in the central brain system [26]. Again, a relatively young (47-year-old) male with IgD myeloma at stage ISS 1 received the first line of VRD treatment and achieved complete remission after 4 cycles. The patient refused the auto-HCT and continued VRD treatment up to 9 cycles followed by the lenalidomide for maintenance. After three months of maintenance therapy, he showed progression in the CNS. The results of blood tests, biochemical detection, serum immunofixation electrophoresis, and bone marrow examination were normal. Ixazomib with lenalidomide and dexamethasone was used as salvage therapy. Radiotherapy was also planned for the occupied places. Unfortunately, the patient died after 1 month due to further systemic progression of the disease.

Crossing of the blood–brain barrier (BBB) by myeloma cells and their growth without engaging the bones or meninges indicates their clonal progression and protects them from the standard agents used for MM treatment, which usually cannot cross the BBB. There is no standard treatment for CNS involvement in MM; however, these patients should receive treatment penetrating CNS (Table 1) [27]. Although appealing, intrathecal triplet therapy (hydrocortisone, methotrexate, and/or cytarabine) is controversial, since studies show only modest benefit [5,11,18]. Radiation remains the treatment of choice and might be combined with systematic treatment. Modern radiotherapy may target the CNS-MM to avoid bone marrow and consequent damage to hematopoiesis. In the study reported by Jurczyszyn et al., patients received systemic treatment (mostly chemotherapy with immunomodulating drugs or proteasome inhibitors for 43% and 33% of patients, respectively) and/or the combination of systemic treatment and radiotherapy. Out of 166 patients receiving treatment, 117 received chemotherapy, 56 had radiotherapy (data regarding the doses not available), and only 1 patient underwent tumor resection. Interestingly, treatment with novel anti-myeloma agents does not result in survival improvement of patients with multiple myeloma and central nervous system involvement [28]. Bendamustine is believed to penetrate the BBB after its administration improved the outcome of two out of three patients with CNS lymphoma [29]. Both topotecan and temozolomide penetrate the BBB, but the evidence of their efficacy in MM treatment is limited [30]. Proteasome inhibitors (PIs) (bortezomib, carfilzomib, and ixazomib) cannot penetrate the BBB [31]; however, bortezomib has been reported to enhance radiosensitivity and chemosensitivity when used in combination with other agents [11,32,33]. There is a report that the novel proteasome inhibitor marizomib can be present in CSF after systemic administration, which suggests its potential activity in CNS-MM [31,32,34]. Immunomodulatory drugs such as thalidomide, lenalidomide, and especially pomalidomide cross the BBB [35]. The presence of thalidomide in the cerebrospinal fluid was found during its use [27,36], but the therapeutic effect was visible only after several weeks, which is not feasible in a disease with a more rapid course. Lenalidomide also crosses the blood–brain barrier but penetrates at a very low level of 5% [11,37]. The most promising of the immunomodulating drugs is pomalidomide, which penetrates the blood–brain barrier at the level of approx. 39% [37]. Zajec et al. performed a study to evaluate the ability of daratumumab to cross the BBB [38]. The concentration of daratumumab in the CSF was approximately 71 times lower than that in serum. According to their study, daratumumab is able to penetrate BBB, but insufficiently, and its concentration is subtherapeutic. However, in cases of possible BBB deterioration, such as an infection or neoplastic disease, the possibility of more efficient daratumumab penetration cannot be excluded [37]. A second auto-HCT might be used as consolidation, as in our case, since melphalan crosses the BBB at high doses [9,27,39]. In the study reported by Jurczyszyn et al., 32 patients (21%) received auto- or allo-HCT after the induction phase.

The new drugs isatuximab, elotuzumab, and venetoclax appear active against myeloma cells but have so far not been used in CNS therapy. Similarly, BITE (bispecific antibody) and CAR-T therapies are still under investigation [26]. Previous studies have not clearly demonstrated the superiority of one therapy over the other. It seems, however, that systemic treatment based on multiple drugs is more effective than single or dual drug therapy (Table 2).

An attractive treatment option for MM invasion of CNS is chimeric antigen receptor-T (CAR-T) therapy which allows for controlling relapsed/refractory MM, particularly B-cell maturation antigen (BCMA)-targeted CAR-T cells [40]. Two BCMA CAR-T therapies are approved by the Federal Drug Agency: Abecma (idecabtagene vicleucel, ide-cel), and Carvykti (ciltacabtagene autoleucel, cilta-cel). The overall response rate (ORR) of these products ranges between 73–97%, and the median PFS time achieved over 8.8 months [41,42,43]. CAR_T cells cross the BBB, which gives a chance for their effectiveness in case of CNS involvement by myeloma cells. However, it was unclear whether indeed CAR-T cells could be effective and also safe in case of CNS involvement. Recently, Wang et al. reported four cases of CNS-MM patients treated with BCMA CAR-T therapy. `Three of them achieved complete response with grades 1–2 cytokine release syndrome (CRS) but without any neurotoxicity, suggesting the safety and feasibility of BCMA CAR-T to treat CNS-MM [44]. One more patient was recently reported by Wang et al. demonstrating the effectiveness and safety of BCMA CAR-T in a CNS-MM patient with multiple other extramedullary lesions. However, it has to be kept in mind that 5% of BMCA-treated patients presented with movement and neurocognitive treatment-emergent adverse events [45] Symptoms were wide-ranging, included movement, cognitive, and personality changes, and occurred after a period of recovery from CRS and/or immune effector cell-associated neurotoxicity syndrome (ICANS) [41].

Prophylactic irradiation of the CNS is not recommended, nor is prophylactic administration of intrathecal chemotherapy or high-dose methotrexate as in other lymphomas because of the lack of specific predictors for the development of myeloma in the central nervous system. That is why maintenance treatment is so important, which aims to reduce the risk of disease progression and extend the time to the next line of treatment.

The prognosis for patients with CNS involvement is rather poor. For the whole group of patients reported by Jurczyszyn et al., median overall survival was 6.7 months. Interestingly, age, sex, and the type of MM had no influence on the duration of survival. Similar results have been presented in other studies over the last 10 years, and the median overall survival of a patient with CNS involvement ranges from 3 to 7.4 months (Table 3). In the paper by Verga et al. in 2018, 13 patients who received systemic treatment (most often three-drug therapy: IMID, proteasome inhibitor, and steroid) with auto-HCT or with radiotherapy had the longest survival time. One patient received daratumumab with good, but short-term, effect [22]. One patient received allotransplantation but died of sepsis.

Although based on the collected literature data and presented by Egan al at., multidirectional therapy including auto-HCT and radiotherapy with or without chemotherapy containing proteasome inhibitors demonstrates the most beneficial effect on the overall survival of patients [1]. Even longer median OS was achieved by 18 patients with CNS-MM who underwent the allo-HCT procedure (25 months) [2].

**Table 2 biomedicines-11-01225-t002:** Overall survival in patients with CNS involvement.

Author	Country	No. of Patient	Year	OS (Months)
Jurczyszyn et al., 2016 [6]	International	172	2000–2015	6.7
Dias et al., 2017 [23]	Brazil	20	2008–2016	7.4
Badros et al., 2016 [34]	USA	2	2008–2016	na
Katodritou et al., 2015 [28]	Greece	31	2000–2013	3
Paludo et al., 2016 [24]	USA, Mayo	29	1998–2014	5
Varga et al., 2018 [22]	Hungary	13	2007–2017	4
Chen et al., 2013 [18]	Canada	37	1999–2010	5
Gozzetti et al., 2012 [46]	Italy	50	2000–2010	6
Abdallah et al., 2014 [4]	USA	35	1996–2012	4

**Table 3 biomedicines-11-01225-t003:** Summary, where data were available. Data from Egan PA, Elder PT, Deighan WI, O’Connor SJM, and Alexander HD. Multiple Myeloma with Central Nervous System Relapse [1].

Median-OS by CNS-MM by Treatment [Months]	CNS-MM Treatment			
	IMID Immunomodulatory Drugs	PIs Proteasome Inhibitors	SCT Stem Cell Transplant	XRT Radiotherapy
5.1	X	X		
4.7	X			X
7.3	X	X		X
5.3			X	
9.0			X	X
6.0		X		X

To conclude, although MM progression to CNS is generally indicative of poor prognosis, our case suggests that patients with CNS involvement without systemic progression may benefit from intensive treatment.

In the last decade, the therapeutic possibilities for myeloma have greatly increased, but we still do not know how to effectively treat patients with CNS involvement, which is why this group of patients requires special observation and reporting of cases. However, the therapeutic options are still very limited. Therapy should be selected for cases individually, depending on the patient’s condition, the presence of comorbidities, and previously used medications. It should contain agents able to penetrate the BBB, and second auto-HCT with a conditioning regimen targeting CNS might be a valid option, since it may significantly prolong survival.

## Figures and Tables

**Figure 1 biomedicines-11-01225-f001:**
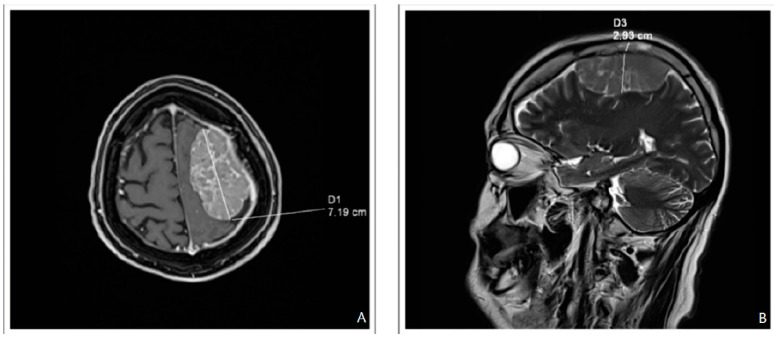
Magnetic resonance imaging (**A**) T1 with contrast, axial view, and (**B**) T2, sagittal view.

**Table 1 biomedicines-11-01225-t001:** Agents used for multiple myeloma treatment and their potential to cross the BBB.

Drug	Crosses the BBB (In Vivo Tests, Laboratory Tests on Animal Patterns, etc.)	Prospective Clinical Trials
Thalidomide	Yes	No
Bortezomib	No	No
Lenalidomide	Yes	No
Pomalidomide	Yes	No
Carfilzomib	No	No
Marizomib	Yes	No
Daratumumab	Yes	No
Bendamustine	Yes	No
Topotecan	Yes	No
Melphalan (high dose)	Yes	No

## Data Availability

The data presented in this study are available on request from the corresponding author. The data are not publicly available due to privacy.

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
