# Peer review of "Isolated Progression of Multiple Myeloma into the Extramedullary Plasmacytoma of Dura Mater: A Case Report and Review of the Literature"

_biomedicines, 2023, doi:10.3390/biomedicines11041225_

Round 1
Reviewer 1 Report
This is a conceptually interesting report. Can the conclusions of the authors be confirmed by more than one case/patient?
Can the authors speculate on consequences for eventual screening strategies in MM patients?
Are there specific FISH / molecular abnormalities associated with this type of presentation ?
Author Response
I am sending the corrected version and checked by english - service. Thank you for your comments.
Reviewer 2 Report
This is a secondary CNS extramedullary plasmacytoma case report and a literature review.
Overall, the case is not novel. However, the authors did an excellent job presenting this case report and reviewing the currently available literature. Unfortunately, we do not have cytogenetics at the presentation.
- Please review the paper for grammar typos.
- Please add selinexor to table 2.
Author Response

(The authors gave the same response as above.)

Reviewer 3 Report
The authors describe a case of multiple myeloma patient with CNS progression which is a very rare but potentially deadly complication for the the patient.
I have only one minor comment - in English, the decimal point is a point, not a comma, please correct in the entire document.
Author Response

(The authors gave the same response as above.)

Round 2
Reviewer 1 Report
The issues of this reviewer have been addressed